# Comparison of the Nutritional Composition of Meat Products Derived from Croatian Indigenous Pig Breeds

**DOI:** 10.3390/foods13244175

**Published:** 2024-12-23

**Authors:** Ana Vulić, Željko Cvetnić, Ivica Kos, Ivan Vnučec, Nada Vahčić, Tina Lešić, Dimitrije Simonović, Nina Kudumija, Jelka Pleadin

**Affiliations:** 1Laboratory for Analytical Chemistry, Croatian Veterinary Institute, Savska Cesta 143, 10000 Zagreb, Croatia; lesic@veinst.hr (T.L.); kudumija@veinst.hr (N.K.); pleadin@veinst.hr (J.P.); 2Croatian Veterinary Institute, Branch–Veterinary Institute Križevci, Ulica Ivana Zakmardija Dijankovečkog 12, 48260 Križevci, Croatia; cvetnic@veinst.hr; 3Faculty of Agriculture, University of Zagreb, Svetošimunska Cesta 25, 10000 Zagreb, Croatia; ikos@agr.hr (I.K.); ivnucec@agr.hr (I.V.); 4Faculty of Food Technology and Biotechnology, University of Zagreb, Pierottijeva 6, 10000 Zagreb, Croatia; nvahcic@pbf.hr; 5Sin Ravnice d.o.o., Stjepana Radića 55, 31531 Cret Viljevski, Croatia; sin_ravnice21@hotmail.com

**Keywords:** dry-cured ham, bacon, dry-fermented sausages, chemical composition, pork

## Abstract

There is a growing interest in the preservation of indigenous pig breeds, as they serve as a valuable genetic reserve. Pork meat products are widely consumed due to their desirable flavor, which is largely influenced by their chemical composition and the production processes employed. The aim of this study was to characterize and compare the nutritional composition, mineral content, and fatty acid profile of meat products derived from indigenous Croatian pig breeds. Three types of meat products, including bacon, dry-cured ham, and dry-fermented sausages, originating from the Turopolje pig, Black Slavonian pig, and Banijska šara, were collected and analyzed for proximate composition, fatty acid profile, and mineral content. Concerning the proximate analysis, statistically significant differences were found in the water and fat content in bacon and dry-fermented sausages, while the mineral analysis revealed differences in iron content. The fatty acid profile of the tested products was found to be in accordance with previously reported data. The results indicated similarities in chemical composition, mineral content, and fatty acid profile between meat products from different pig breeds; however, performing PCA analysis revealed that the major influence on product and breed characterization could be attributed to differences in fatty acid composition.

## 1. Introduction

Meat and various meat products derived from pork represent an essential component of the diet for the global population, while currently, there are approximately 400 local pig breeds worldwide [1]. In the past, the increasing demand for pork and the subsequent intensification of production have contributed to a decline in indigenous pig breeds, which are usually bred extensively and have lower yield. Literature data indicate that there is a growing interest in the preservation of indigenous pig breeds, as they serve as valuable genetic reserves [2]. In Croatia, three indigenous pig breeds can be identified: the Turopolje pig (Turopoljska svinja), the Black Slavonian pig (Crna slavonska svinja), and the Banijska šara [3]. The Turopolje pig, which has its origins in the sixth century, was developed through the crossbreeding of the Šiška (Bosnian primitive pig) and the Slovenian Krškopolje pig. This breed is categorized as a fatty type, and its meat is characterized by its juiciness and tenderness. Notably, the meat of the Turopolje pig exhibits a darker and redder color than conventional pork, along with a fine muscular texture that is distinctive of this breed [4]. It is predominantly reared in the Turopolje region. The Black Slavonian pig (Crna slavonska), which emerged during the nineteenth century, resulted from the planned crossbreeding of four different breeds: Mangalitsa, Berkshire, Poland China, and Large Black. This breed is classified as a fatty-meat type and is primarily utilized for the production of traditional pork products, including ham, kulen, bacon, sausages, and dry-cured neck [5]. It is reared across several counties, including Brod-Posavina, Požega-Slavonia, Osijek-Baranja, Vukovar-Srijem, and Sisak-Moslavina. Due to the distinct characteristics associated with their rearing areas, the meat of both the Turopolje pig and Black Slavonian pig is protected under the designation of Protected Designation of Origin (PDO). The Banijska šara is a lesser-known indigenous pig breed in Croatia that was developed through the crossbreeding of local pigs with Berkshire. This breed falls within the fatty type category and is predominantly reared in Sisak-Moslavina County [6]. All the aforementioned pig breeds are raised extensively and are primarily used in the preparation of various meat products, including bacon, ham, and sausages.

Pork meat products are widely consumed due to their desirable flavor, which is largely influenced by their chemical composition and the production processes employed. Specifically, the high levels of proteins and fats, combined with aromatic compounds formed during fermentation or drying, contribute to the appealing flavor of these products [7]. Although variations in recipes and production techniques result in significant differences among specific types of meat products, certain variations can also be attributed to the breed of the pig. Current trends in nutrition indicate a notable shift towards reducing meat consumption, particularly red meat, motivated by health considerations and environmental concerns linked to the intensive breeding practices of meat-producing animals. This reduction aims to promote potential health benefits and has a positive impact on efforts to mitigate climate change. Furthermore, consumers are increasingly focused on the purchase of value-added meat products, which include traditional meat offerings, products resulting from ecological farming, and those distinguished by health claims, as well as products protected as Protected Designation of Origin (PDO), Protected Geographical Indication (PGI), and Traditional Specialty Guaranteed (TSG) [8]. Native pig breeds are contributing to this trend, as the organoleptic properties of pig meat were partially lost due to the selective breeding programs that were aiming to increase pig production [2].

The aim of this study was to characterize and compare the nutritional composition, mineral content, and fatty acid (FA) profile of pork products (bacon, dry-cured ham, and dry-fermented sausages) derived from indigenous Croatian pig breeds, each characterized by specificities in rearing regimes. The study included three types of meat products: bacon, dry-cured ham, and dry-fermented sausages, which were produced solely in households registered to breed indigenous pig breeds.

## 2. Materials and Methods

### 2.1. Animals and Meat Products

Three types of meat products, including bacon, dry-cured ham, and dry-fermented sausages, originating from Turopolje pig (TP), Black Slavonian pig (BSP), and Banijska šara (BŠ) were collected in family households located in the Central and Eastern part of Croatia. In total, 18 bacon samples (7 TP, 5 BSP, 6 BŠ), 16 dry-cured hams (6 TP, 5 BSP, 5 BŠ), and 21 dry-fermented sausages (9 TP, 6 BSP, 6 BŠ) were collected. The number of samples was limited due to the number of households that are breeding indigenous pig breeds but also producing meat products. All pigs used for the production of meat products were male castrated fattening pigs slaughtered at the age of 12 months for BS, 16 months for BSP, and 20 months for TS.

### 2.2. Production of Bacon, Dry-Cured Ham, and Dry-Fermented Sausages

Bacon was made from pork belly, which was first salted at temperatures between 2 °C and 5 °C, then smoked at temperatures below 22 °C, then dried at temperatures between 10 °C and 20 °C (with a relative humidity of 65% to 75%), and in the end, cured at temperatures between 8 °C and 18 °C, as previously described [9,10]. In addition to bacon, the dry-cured ham, which is made from pork legs, was produced. The dry-cured hams were first salted at low temperatures between 2 °C and 5 °C, then massaged and cured. The curing step included smoking at temperatures below 22 °C, drying at temperatures between 10 °C and 20 °C (with a relative humidity of 65% to 75%), and curing at temperatures between 8 °C and 18 °C. The dry-fermented sausages were produced solely from pork and pork fat with the addition of salt, black pepper, and dried red paprika. The filling was then stuffed into casings and subjected to the processes of drying, fermentation, and maturation according to the previously described procedures [11,12].

### 2.3. Reagents and Standards

Petrol ether, hydrochloric acid, sulphuric acid, sodium hydroxide, sodium chloride, and boric acid were from Sigma (USA). Hexane and isooctane were from Merck (Darmstadt, Germany). A standard solution of fatty acid methyl esters (FAMEs) was prepared by dissolving 100 mg of standard SupelcoTM 37 Component FAME Mix (Supelco, Bellefonte, PA, USA) in 10 mL of hexane. Ultra-pure water with an electrolytic conductivity of ≤0.05 S/cm was obtained using Milipore Direct-Q 3UV (Merck, Germany). The standard solution for each mineral tested (1000 μg/mL) in 5% nitric acid was provided by Agilent Technologies (Santa Clara, CA, USA).

### 2.4. The Proximate Analysis

Compositional analysis (g/100 g) was performed by applying validated standard and internal analytical methods. The determination of water [13] and ash [14] content was performed by gravimetric methods with the use of a thermostat (UF75 plus, Memmert, Germany) and muffler burning furnace (Program Controller LV 9/11/P320, Nabertherm, Germany). The crude protein content was determined by the Kjeldahl method, according to standards [15,16], which involves the destruction of organic matter at 420 °C with the use of a block digestion unit (Unit 8 Basic, Foss, Hilleroed, Denmark) combined with a titration and distillation unit (Vapodest 50s, Gerhardt, Germany). The crude fat content was determined by the Soxhlet method [17], which includes fat hydrolysis and fat extraction with petrol ether on the extraction device (Soxtherm 2000, Gerhardt, Germany). The carbohydrate content was calculated by subtracting protein, fat, ash, and water content from 100. The sugar content was established using a spectrophotometer (DR 6000, HACH, Lange, Iowa City, IA, USA) and a commercial enzyme kit (Enzytec liquid sucrose/D-Glucose/D-Fructose, R-Biopharm, Pfungstadt, Germany) according to the instructions by the kit manufacturer. The salt content was determined stoichiometrically based on the established sodium content determined by an Atomic Absorption Spectroscopy (AAS) method, together with other minerals.

### 2.5. Fatty Acid Methyl Esters (FAMEs) Analysis

Fat obtained during proximate analysis was used for the preparation of fatty acid methyl esters (FAMEs) following the standard method [18] with some modifications. Briefly, sixty milligrams of the extracted fat was dissolved in 4 mL of isooctane, followed by the addition of 200 μL of 2 N methanolic potassium hydroxide solution. The samples were then shaken for 60 s. Subsequently, 4 mL of saturated sodium chloride solution (300 g/L) was added, and the samples were vortexed and left at room temperature to allow layer separation. The upper isooctane layer was transferred to another test tube, where 2 g of anhydrous sodium hydrogen sulfate was added. The samples were then centrifuged for 15 min at 3000 rpm and 15 °C. Two hundred microliters of each sample was filtered through a PTFE filter (0.2 µm pore size) into vials for analysis. The FAMEs were analyzed using a gas chromatography–flame ionization detection (GC-FID) method with a 7890 A gas chromatograph (Agilent Technologies, USA) equipped with a DB-23 capillary column (60 m length, 0.25 mm internal diameter, and 0.20 µm stationary phase thickness; Agilent Technologies, USA). Helium was used as the carrier gas in constant flow mode at 2 mL/min. The oven temperature program was as follows: an initial temperature of 120° C was maintained for 1 min, increased at a rate of 10 °C/min to 175 °C, held for 10 min, increased at a rate of 5 °C/min to 210 °C, held for 5 min, and then increased at a rate of 5 °C/min to a final temperature of 230 °C, which was maintained for 5 min. The FID detector was set at 280 °C, with hydrogen, air, and nitrogen flows set at 40 mL/min, 450 mL/min, and 30 mL/min, respectively. The split/splitless injector was set at 250 °C, and one microliter of the sample was injected with a split ratio of 1:50. The identification of FAMEs was achieved by comparing the retention times of FAMEs in the sample with those of a standard solution mixture. Given the modifications to the standard method, quality control was ensured using the CRM BCR163 reference material (Institute for Reference Materials and Measurements, Belgium). The content of seven individual fatty acids in the reference material was compared with certified values and assigned tolerances to validate the method.

### 2.6. Fat Quality Indices

The lipid quality indices, including atherogenicity index (IA), thrombogenicity index (IT), and the ratio of hypocholesterolemic and hypercholesterolemic fatty acids (HH), were calculated according to the following equations, given by [19,20].
IA = (C12:0 + 4 × (C14:0) + PALMITIC ACID (C16:0))/MUFA + PUFA(1)
IT = C14:0 + PALMITIC ACID (C16:0) + C18:0/(0.5*MUFA + 0.5 × n6PUFA + 3 × n3PUFA + n3/n6)(2)
HH = C18:1n9 + C18:2n6 + C20:4n6 + C18:3n3 + C20:5n3 + C22:5n3 + C22:6n3/C14:0 + palmitic acid (C16:0)(3)
where MUFA and PUFA stand for monounsaturated and polyunsaturated fatty acids, respectively.

### 2.7. Mineral Analysis

The in-house AAS method was used for mineral content determination. Briefly, the sample was digested in the presence of nitric acid and hydrogen peroxide in a microwave oven (Ethos easy, Milestone, Italy). Afterwards, samples were diluted with ultrapure water in a volumetric flask, followed by measurement of sodium (Na), calcium (Ca), potassium (K), magnesium (Mg), copper (Cu), zinc (Zn), and iron (Fe) by flame AAS (200 Series A4 with SPS 4 Autosampler, Agilent Technologies, USA). For each mineral, an HC-coded lamp specific to the given mineral (Agilent Technologies, USA) was used. The procedure was described in detail earlier [9].

### 2.8. Statistical Analysis

Statistical analyses were performed using SPSS Statistics Software 22.0 (IBM, New York, NY, USA). The differences between the three pig breeds in the analyzed parameters were established using the analysis of variance (ANOVA) with either the Tamhane’s T2 or the Scheffe post hoc test, depending on the variance analyzed by the Levene test, with the statistical significance set at 95% (*p* < 0.05). The results were submitted to Principal Component Analysis (PCA) in order to interpret the chemical composition, mineral content, and fatty acid profile among pig breeds using the Statistica 10.0 Software (StatSoft, Palo Alto, CA, USA).

## 3. Results and Discussion

### 3.1. Nutritive Characteristics

The results of the proximate analysis are shown in Figure 1. In bacon samples, as expected, fat had the highest share, ranging from 62.72 g/100 g for Banijska šara up to 77.46 g/100 g for the Turopolje pig. The content of other macronutrients, protein, and carbohydrates ranged from 9.25 g/100 g up to 12.68 g/100 g and 1.58 g/100 g up to 4.36 g/100 g, respectively. The protein, ash, salt, carbohydrate, and sugar content in the bacon samples was constant for all pig breeds, while statistically significant differences were found in the water and fat content. For water content, a significant difference (*p* = 0.008) was found between Banijska šara and the Turopolje pig, while the fat content in the bacon samples from the Turopolje pig was statistically different (*p* = 0.006) from both the Black Slavonian pig and Banijska šara.

Dry-cured hams originating from Croatian native pig breeds were also analyzed in the presented study. The macronutrient content, meaning the share of protein, fat, and carbohydrates, in the Turopolje pig, Black Slavonian pig, and Banijska šara ranged from 24.57 to 27.64 g/100 g, 21.72 to 38.24 g/100 g, and 0.95 to 3.02 g/100 g, respectively. In this group of tested samples, there was no statistically significant difference between pig breeds regarding any analyzed parameter.

The last group of meat products in the study were dry-fermented sausages. This type of meat product is, in contrast to bacon and dry-cured hams, produced from several ingredients and spices, as well as according to different recaptures, so more differences among groups were expected. The results revealed that there was a statistically significant difference in water content (*p* = 0.015), as well as in fat content (*p* = 0.033), between Turopolje pig and Banijska šara sausages. The share of protein, fat, and carbohydrates in Turopolje pig, Black Slavonian pig, and Banijska šara sausages ranged from 28.84 to 30.41 g/100 g, 36.26 to 45.00 g/100 g, and 1.15 to 1.48 g/100 g, respectively. When resuming the proximate analysis, the salt content among analyzed meat products must be addressed. The current trends concerning salt intake point to the need to reduce salt content in meat products [21]. The results of salt analysis of samples in this study revealed that the lowest salt content can be attributed to the bacon samples (around 3 g/100 g), followed by dry-fermented sausages (3.5–4.5 g/100 g), while the highest share was determined to be in dry-cured hams (5.5–6.5 g/100 g).

To date, there have been numerous studies that have assessed the chemical composition of traditional meat products in several European countries, including Croatia, but only a couple of them in meat products originating from native pig breeds [22,23,24]. When comparing the results from the presented study with previous research on Croatian dry-fermented sausages [9,25,26,27], it can be observed that there is a similarity in chemical composition. In the mentioned studies, the protein content spans from 25 to 30 g/100 g, the fat content is about 30–40 g/100 g, the water content is 25–35 g/100 g, the carbohydrate content is 0.5–1.0 g/100 g, and the salt content is 3.5–4.5 g/100 g. The relatively wide range of the individual components can be linked to the variety of recipes that are used for the production of this type of meat product. In contrast with these findings, the results of the nutritive characterization of the Spanish traditional sausage, Asturian Chorizo [28], revealed much higher fat content (more than 55 g/100 g) and, consequently, lower water content (around 33 g/100 g). Similar results were presented in the study by Rason et al. [29], also dealing with traditional sausages but originating from France. The study included an analysis of the same types of sausages during a three-year period, and the results also revealed a high fat content (up to 56 g/100 g). Results from the characterization of traditional Portuguese sausages [30], as well as traditional Greek sausages [31], revealed a higher diversity of nutritional components. The fat content spans from 10 to 45 g/100 g and 12 to 53 g/100 g, while the protein content spans from 13 to 26 g/100 g and 11 to 30 g/100 g in Greek and Portuguese traditional sausages, respectively. Dry-cured ham is a meat product that is produced from the pork leg, and it does not include additional ingredients, except salt that is used for curing. Bearing this in mind, the variations in nutritive composition are less expected compared to the dry-fermented sausages. The production of dry-cured hams is characteristic of several Mediterranean countries, i.e., Portugal, Spain, Italy, Slovenia, Croatia, Serbia, Bosnia and Herzegovina, Montenegro, and Greece. When comparing the present results with the study that embraced several dry-cured hams originating from different parts of Croatia [32], differences can be observed concerning protein and fat, while there is a similarity in the salt content. The dry-cured hams from the presented study had lower protein and higher fat content compared to the study from [32]. Concerning the salt content, values above 5 g/100 g were determined in samples from both studies, indicating that these products can be categorized as products with high salt content.

### 3.2. Minerals

Meat and meat products are considered an excellent source of macro- and micro-minerals in the diet. The mineral content in the muscle of food-producing animals depends on the breed, species, type of meat, diet, heat treatment, and sampling methods [33]. The mineral composition of the analyzed meat products is shown in Table 1. Calcium is the most abundant mineral in the body, and the body needs it for numerous functions, e.g., for vascular contraction, vasodilation, muscle function, oocyte activation, blood clotting, nerve transmission, intracellular signaling, hormone secretion in the human body, regulation of the heartbeat, and regulation of fluid balance in cells [34]. The calcium requirement for adults is estimated at 700–1300 mg per day [35]. The results of the present study showed an average calcium content between 200 mg/kg and 300 mg/kg. These results indicate that this type of meat product could be an adequate source of calcium when considering the daily amount required. When comparing the results for the same type of meat products between different breeds of pigs, no statistically significant difference in calcium content was found.

Phosphorus is a mineral that is an important component of bone and is essential for several functions in the organism, including bone mineralization, energy production, kidney function, cell growth/signaling, and regulation of acid–base homeostasis [33,35]. The daily requirement of phosphorus for adults is between 700 and 1300 mg per day, and deficiency is rather rare. The dry-fermented sausages from our study had the highest phosphorus content at around 0.25 g/100 g, followed by the dry-cured ham and bacon samples, although there were no statistically significant differences in phosphorus content between the products of the different pig breeds.

Another important mineral for the human body is potassium. Together with sodium, it plays an important role in the regulation of electrolytes, blood pressure, insulin secretion, creatine phosphorylation, carbohydrate metabolism, protein synthesis, and as a co-factor for many enzymes [33]. The potassium levels in the analyzed sample groups were very different, with the highest levels found in the dry-fermented sausages from the Turopolje pig and the lowest in the bacon sample from the Black Slavonian pig, but there was no significant difference between the meat products of the different pig breeds.

The last mineral analyzed from the group of macro-minerals was magnesium. Its function is associated with metabolic processes in the body, including energy production, cell growth, and the synthesis of biomolecules [33,36]. As with the other macro-minerals, there were no significant differences in magnesium content between the products of the different pig breeds. The magnesium content was highest in dry-fermented sausages, followed by dry-cured ham, while the lowest average content was found in bacon samples.

Iron, copper, and zinc are minerals that belong to the group of micro-minerals. Iron is an important mineral for human health, as it plays a crucial role in numerous metabolic processes, including the synthesis of deoxyribonucleic acid (DNA), oxygen transport, nutrient transport, and the formation of hem enzymes and other iron-containing enzymes [33]. Interestingly, the only significant difference between all analyzed minerals was found in the iron content in dry-fermented sausages, with the Black Slavonian pig differing from both the Turopolje pig and Banijska šara (*p* = 0.036). Iron deficiency can lead to anemia and lower immunity to infections. The recommended intake of iron is 8–18 mg per day, depending on age and sex [37]. When comparing the results with these values, it can be concluded that the consumption of around 200 g of dry-fermented sausage (Turopolje pig or Banijska šara) could provide the minimum recommended daily value for iron. The research focused on the mineral composition of bacon and dry-cured ham, but there have been several studies looking at the minerals in various dry-fermented sausages. The study by Marcos et al. [30] assessed the nutritive value of selected Portuguese sausages. Concerning the variety of tested sausages, the differences in mineral content were expected, but quite a discordance in iron content was observed. The iron content in the samples in our study had at least 10-fold higher iron concentration. Also, in the study from Kudumija et al. [9], the iron concentration was at the same level as in sausage samples from the Black Slavonian pig but was almost three times lower than in samples from the Turopolje pig and Banijska šara.

### 3.3. Fatty Acid Profile

Table 2 shows the fatty acid profile of the analyzed products. Fatty acids are part of the human diet but can also be found in human cells and tissues. Fatty acids are used for energy, and they are used to build cell walls. They have different biological activities, and through these biological actions, fatty acids influence health, well-being, and disease risk. The fatty acid (FA) profile determined in bacon revealed that oleic acid (C18:1n-9c) is the predominant FA in the whole group, as well as in the group of MUFAs. Its amount ranged from 35.57% up to 44.06%, but there was no significant difference among pig breeds. In the group of saturated FAs, palmitic acid (C16:0) had the highest share, ranging from 23.78% up to 24.09%, while in the group of PUFAs, the predominant PUFA was linoleic acid (C18:2n-6c), ranging from 8.12% up to 9.27%. The only statistically significant difference (*p* = 0.040) was determined for the eicosatrienoic acid (C20:3n-3) FA between bacon originating from the Black Slavonian pig and Banijska šara. Interestingly, the low levels of eicosapentaenoic acid (C20:5n-3) were found only in bacon samples but in all three breeds tested. The results of FA analysis in dry-cured hams followed the same pattern as bacon samples, with oleic acid (C18:1n-9c) being the predominant FA, with the share ranging from 35.14% up to 43.33%. When comparing the other classes of FAs between bacon and dry-cured ham samples, the highest share of SFAs can be attributed to palmitic acid (C16:0) (23.72–28.15%), and in the class of PUFAs, to linoleic acid (C18:2n-6c) (7.13–8.23%). The only statistically significant difference was determined for the eicosatrienoic acid (C20:3n-3) and dihomo-gamma-linolenic acid (C20:3n-6) FAs between the Black Slavonian pig and Banijska šara (*p* = 0.005 and *p* = 0.039, respectively). The results of FA analysis in the group of dry-fermented sausages are comparable with the previous groups of meat products. The predominant FA was oleic acid (C18:1n-9c) (34.77–42.92%), while in the class of SFAs and PUFAs, palmitic acid (C16:0) (23.95–25.39%) and linoleic acid (C18:2n-6c) (7.31–9.58%) FAs had the highest share. Dry-fermented sausages are, in contrast to bacon and dry-cured ham, produced from several ingredients and added spices, which results in larger diversity in the FA profile. As a result, the FA profile of dry-fermented sausages originating from different pig breeds has shown statistical differences for many FAs (Table 2).

The production of traditional meat products, i.e., dry-cured hams and dry-fermented sausages, is common in many European countries, including Croatia [32,38]. Our previous research also assessed the FA profiles among dry-cured hams, dry-fermented sausages, and other types of traditional meat products but without addressing the meat origin and pig breed [9,25,27,38]. The results of FA analysis presented in this study are in accordance with our previous research revealing that among PUFAs, oleic acid (C18:1n-9c) is the predominant FA, palmitic acid (C16:0) is predominant in the group of SFAs, and linoleic acid (C18:2n-6c) is predominant in the group of PUFAs. These findings support the fact that the profile of fatty acids and the proportion of the most-represented FAs are consistent and do not depend on the origin of the pork meat nor on the breed of the pig, as demonstrated in the present study. Several studies were conducted by Spanish and Italian research groups and were focused on dry-cured hams characteristic of that country [39,40,41,42]. Concerning the FA profile, these studies also revealed that the profile of fatty acids and the proportion of the most-represented FAs are consistent among the varieties of dry-cured hams. Concerning the FA profile in bacon samples, the literature data are limited. As the Black Slavonian pig is a native Croatian pig, some research was implemented to compare this pig breed with modern ones. This study pointed to a significant difference in several FAs, including capric acid (C10:0), lauric acid (C12:0), palmitic acid (C16:0), stearic acid (C18:0), and arachidic acid (C20:0) in the class of SFAs and palmitoleic acid (C16:1), heptadecenoic acid (C17:1), oleic acid (C18:1n-9t), and eicosenoic acid (C20:1) in the class of MUFAs [43]. Data revealed in our study are in line with the study from Latin et al. [43] regarding the fatty acid profile.

### 3.4. Lipid Quality Indices

In Table 3, fat quality indices for bacon, dry-cured ham, and dry-fermented sausages are presented. Several indices are generally recognized for the assessment of fat quality, including the ratio of omega-6 and omega-3 fatty acids, the ratio of PUFAs and SFAs, the ratio of hypo-/hypercholesterolemic fatty acids (H/H index), the atherogenic index (IA), and the thrombogenic index (IT) [44,45]. The PUFA/SFA index is used to assess the impact of diet on cardiovascular health and the nutritional value of food and should be above 0.4 [46]. The PUFA/SFA indices for the meat products analyzed in the present study ranged from 0.22 to 0.31, with the highest value found in the bacon samples from Banijska šara. None of the samples analyzed had an FA profile resulting in a PUFA/SFA ratio that met favorable values. Similar results were presented in the study by Wójciak et al. [47], in which the PUFA/SFA ratio for ham samples was between 0.21 ± 0.04 and 0.26 ± 0.04. Another study evaluating the quality of the fat and FA profile was presented by Fernández et al. [39] and pointed to the same results in relation to the PUFA to SFA ratio, which ranged from 0.19 to 0.30. The IA indicates the ratio between the sum of saturated fatty acids (SFAs) and the sum of unsaturated fatty acids (UFAs). The saturated fatty acids, including lauric acid (C12:0), myristic acid (C14:0), and palmitic acid (C16:0), with the exception of stearic acid (C18:0), are considered pro-atherogenic and favor the adhesion of lipids to cells. The favorable value for IA is below 1. The results of the analysis of all samples of bacon, dry-cured ham, and dry-fermented sausage in the study showed an IA below 1, regardless of the pig breed. The IA values ranged from 0.46 for dry-cured ham from Black Slavonian pigs to a maximum value of 0.72 for dry-cured ham from Turopolje pigs. Similar results were obtained in the studies by Wójciak et al. [47] and Campo et al. [40], which focused on dried ham produced in Poland and Spain. The IAs in both studies corresponded to favorable values, with values of 0.50 ± 0.04 and between 0.37 ± 0.03 and 0.41 ± 0.04 in these studies, respectively. Previous studies evaluating the IA for dry-fermented sausages followed the same pattern as for dry-cured ham. In the study on two types of dry-fermented sausages, the determined IA values were below 1 for both types of sausage [27]. The study by Lešić et al. [38], which examined five different traditional dry-fermented sausages, also showed IA values below 1. Another index used to assess fat quality is IT, which characterizes the thrombogenic potential of FAs, i.e., the ability to form clots in blood vessels. Different FAs contribute differently to this process, so there are the pro-thrombogenic FAs (lauric acid (C12:0), myristic acid (C14:0), and palmitic acid (C16:0)) and the anti-thrombogenic FAs (MUFAs and the n-3 and n-6 FA families). Like IA, the favorable value for IT is below 1, but the results of various studies, including this study, indicate that pork products do not meet this value. All ITs calculated in this study were above 1 and ranged from 1.08 ± 0.18 for bacon from Black Slavonian pigs to 1.69 ± 1.23 for dry-cured ham from Turopolje pigs. Similar results were presented for the ham samples in the study by Wójciak et al. [47], with IT values ranging from 1.10 ± 0.11 to 1.15 ± 0.11. Interestingly, in a study on traditional Spanish ham, much higher IT values were found, with IT values ranging from 5.29 ± 1.52 to 8.09 ± 1.70. These results could be attributed to the significantly higher proportion of oleic acid (C18:1 n-9 FA) considered in the calculation of IA in Spanish dry-cured ham (> 45%) compared to the present study, where oleic acid ranged from 34.77% to 44.06%. As for the dry-fermented sausages, the ITs reported by other researchers were in agreement with our study. Lešić et al. [38] reported ITs between 1.22 and 1.43 in five types of traditional Croatian dry-fermented sausages, while Pleadin et al. [27] conducted a study on two types of Croatian dry-fermented sausages with ITs around 0.46. The HH index characterizes the ratio between hypocholesterolemic (cis-C18:1 and PUFA) and hypercholesterolemic FAs, and to ensure a positive impact on health, it is desirable that this index is as high as possible. Our results showed that the highest HH index was calculated for Black Slavonian pork dry-cured ham (2.17 ± 0.26) and the lowest for Banijska šara dry-cured ham (1.67 ± 0.80). Similar to our results, the study from Fernández et al. [39] reported HH indices between 2.00 and 2.67 for traditional Spanish hams. As for dry-fermented sausages, previous results from the studies by Pleadin et al. [27] and Lešić et al. [38] reported HH index values around or slightly above 2. The literature data show that a favorable n-6/n-3 ratio should preferably be between 1/1 and 4/1 [39], which is not the case for the samples tested in our study. The n-6/n-3 ratio calculated for bacon, dry-cured ham, and dry-cured sausages ranged from 18.71 ± 5.60 to 27.15 ± 8.05, with the lowest value calculated for bacon from Banijska šara and the highest value for dry-cured ham from Turopolje pigs. These results are not consistent with the study by Wójciak et al. [48], where the analysis of FAs in ham samples showed an n-6/n-3 ratio between 8.57 ± 1.61 and 9.64 ± 0.84. When comparing these two studies, the n-6/n-3 ratio was twice or three times higher in the samples from Croatian native pig breeds. The different results of the two compared studies could be explained by the fact that indigenous pig breeds accumulate higher amounts of fat than modern breeds, with higher amounts of SFAs and PUFAs [49].

### 3.5. Principal Component Analysis (PCA)

Principal Component Analysis (PCA) is a statistical technique used to reduce the dimensionality of large datasets by identifying the most important components that capture the majority of variance in the data. In this research, PCA was applied to assess the relationships between various quality traits, including basic chemical composition, mineral profile, and fatty acid composition, while distinguishing product samples from different breeds. The first two principal components (PC1 and PC2) explained 40.18% of the variance in the dataset (PC1 20.03%; PC2 18.15%). As seen in Figure 2, PC1 was positively associated with saturated FAs and MUFAs containing fewer than 16 carbon atoms, as well as the atherogenic index, sodium, manganese, iron, salt, and ash content, suggesting that increases in these traits push the data along the positive direction of PC1. On the other hand, PUFAs, especially omega-6, contributed negatively to PC1, indicating that samples with higher PUFA content tend to score lower on PC1. In contrast, PC2 was predominantly influenced by MUFAs, particularly C20:1, and trans fatty acids (TFAs) on the positive side, while saturated fatty acids (SFAs) and the thrombogenic index (TI), as well as C18:0 and C18:1 fatty acids, had the strongest negative impact. Overall, while the basic chemical composition and mineral content showed a low contribution to PCA loadings, the fatty acid profile—particularly specific fatty acids such as C20:2n6 and C18:2n6 on PC1, as well as C18:0, C18:1, C20:1, and C18:3n6 on PC2—along with salt and sodium content, had the highest influence on the distribution of samples within the PCA space.

The scores plot of the samples in the PCA space, defined by the first two principal components, is presented in Figure 3. It was found that products originating from Turopolje pigs (marked in green) were more distributed along PC2, with greater concentration in the middle of the PCA plane and a higher presence in the right quadrants, where saturated fats contributed the most. All product types were dispersed, with the highest dispersion observed in dry bacon samples (marked with circles), followed by ham (marked with triangles) and sausage (marked with squares) samples. Dry meat products from Banijska šara pigs (marked in red) appeared to be the most dispersed across all four quadrants of the PCA plane. Notably, dry ham samples were mostly located in the right quadrants, where saturated fatty acids, sodium, and salt made the greatest contributions. Compared to other pig breeds, dry bacon and sausage samples from Banijska šara were more frequently placed in the left quadrants, where PUFAs had the highest contribution. Dry meat products from Black Slavonian pigs (marked in orange) were primarily located in the lower quadrants, in areas where the most abundant fatty acid, C18:1, along with saturated fatty acids (SFAs), had the greatest contribution. Bacon samples tended to be positioned more to the left, while ham and sausage samples were more frequently placed to the right.

Performing PCA analysis revealed that the major influence on product and breed characterization could be attributed to differences in fatty acid composition. Considering the variability found in previous studies [23], these results were expected and further confirmed. When taking into account the different production strategies of these pig breeds [50], as well as varying management and feeding practices among producers of the same breed, the variability in meat products is highly probable and confirmed in this research. Furthermore, it is important to highlight that manufacturing procedures, particularly those resulting in different salt or sodium contents, represent another significant source of variation in dry meat products from native pig breeds. This also extends to noticeable differences in recipes and production duration, during which extensive changes in fatty acid and amino acid composition occur, as stated by several studies [51,52,53,54]. Although the richness of specific and unique flavors typically found in meat products from native pig breeds [24,55] indicates high market differentiation potential, the variability within each breed origin temporarily illustrates a significant need for standardization.

In summarizing the observed differences in chemical composition, iron content, and fatty acid profile, several issues must be addressed. Differences in water content among groups of bacon samples and dry-fermented sausages could be attributed not only to the pig breed but also to the duration of the drying process, which ranged from 70 to 90 days for bacon and 60 to 75 days for dry-fermented sausages. The variations in fat content among the dry-fermented sausages may result from minor differences in applied recipes, whereas for bacon samples, the differences could be attributed to variations in cuts of meat. Regarding fat content, variations in the FA profile among dry-fermented sausages warrant attention, but as the FAs are constituents of fat, the differences could be attributed to the differences in fat content. The type of feed is also important; the pig is an animal that does not substantially modify the fat consumed during the digestion of the feed, and it deposits this fat in the tissues with little or no modification [56]. The result is that the FA profile of the lipid deposits is more or less the same as the fat found in the feed. Iron content also varied, but only within the group of dry-fermented sausages. Since this group of meat products differs in recipes concerning the amount of paprika used, this could be another possible source of variation, in addition to the pig breed. Furthermore, the age and gender of the animal play important roles in chemical composition and fatty acid profiles. As noted in the study by Zomeño et al. [57], gender affects fat, protein, and moisture content. The same study also found that meat from castrated male pigs contained higher levels of saturated fatty acids (SFAs) and monounsaturated fatty acids (MUFAs) compared to meat from females or chemically castrated male pigs. Similar findings were reported by Teye [58], who indicated that increasing age and weight resulted in higher concentrations of SFAs while significantly lowering the concentrations of PUFAs.

## 4. Conclusions

The presented study contributes to the characterization and standardization of several meat products produced from the meat of Croatian indigenous pig breeds, which have been highly appreciated by consumers over the last decade. The proximate analysis revealed statistically significant differences in water and fat content among bacon and dry-fermented sausages from different pig breeds. Among all minerals tested, the only difference observed was in iron content. The results indicated similarities in chemical composition, mineral content, and fatty acid profile between meat products from different pig breeds; however, performing PCA analysis revealed that the major influence on product and breed characterization could be attributed to differences in fatty acid composition. These results indicate that there are no significant effects on the nutritional composition of meat products despite different husbandry conditions and genetic characteristics of different pig breeds.

## Figures and Tables

**Figure 1 foods-13-04175-f001:**
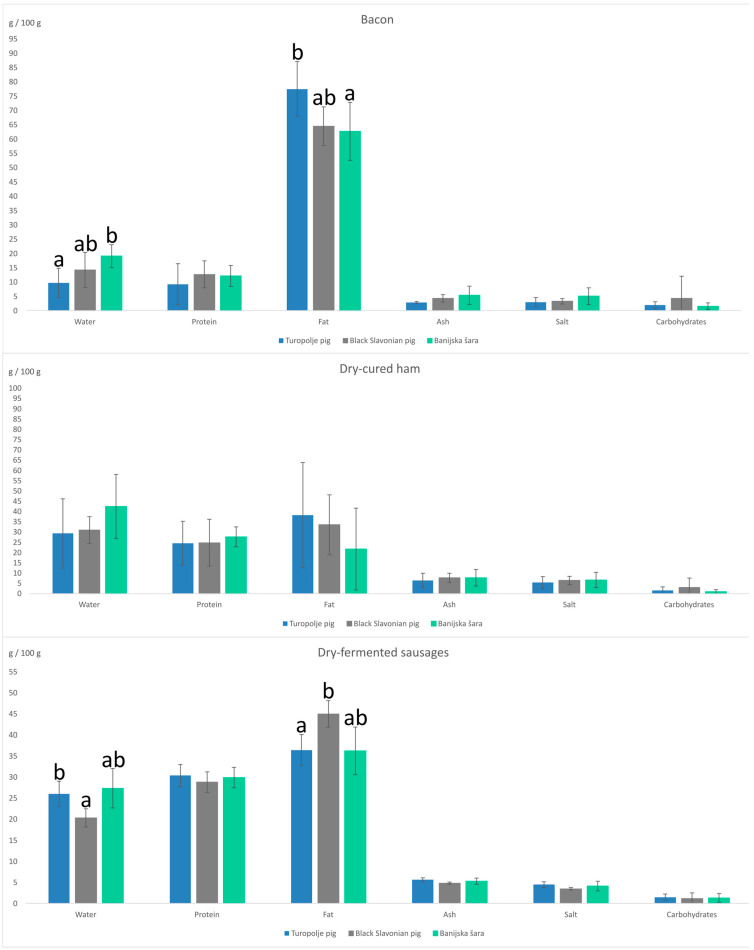
The results of the proximate analysis of bacon, dry-cured ham, and dry-fermented sausages originating from Croatian indigenous pig breeds. a,b: values with no common superscript differ significantly (*p* < 0.05).

**Figure 2 foods-13-04175-f002:**
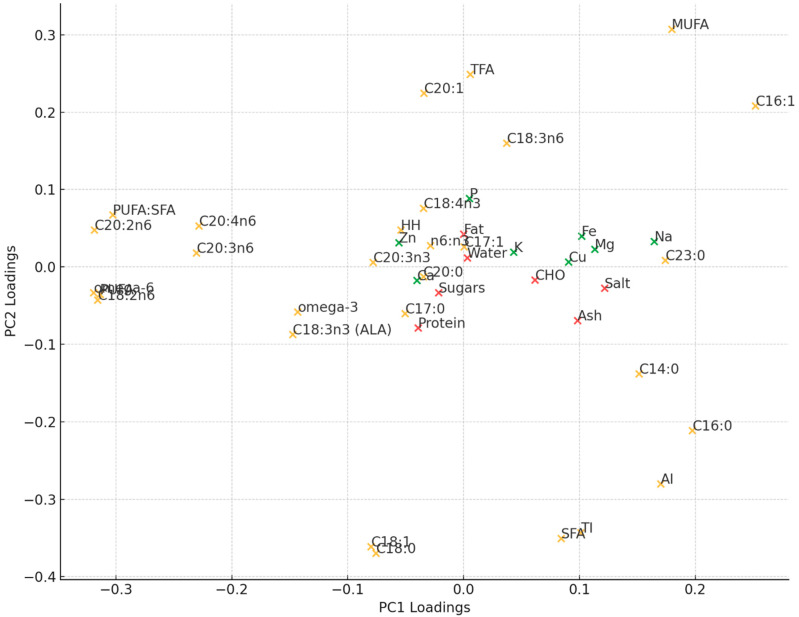
Loading plot of variables on first two principal components.

**Figure 3 foods-13-04175-f003:**
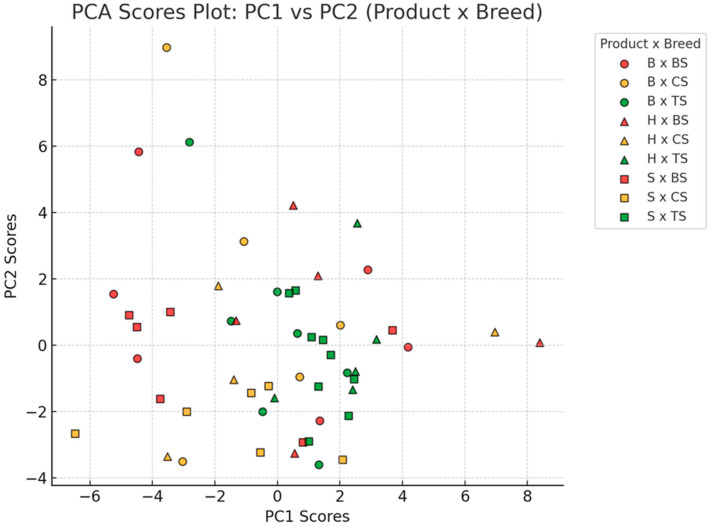
Scores plot of the samples in PCA space defined with first two principal components (B—bacon; H—dry ham; S—dry sausages; BS—Banijska šara; CS—Black Slavonian pig; TS—Turopolje pig).

**Table 1 foods-13-04175-t001:** Mineral composition of different meat products originating from Croatian indigenous pig breeds.

Meat Product	Pig Breed	Mineral							
		Ca (mg/kg ± SD)	P (g/100 g ± SD)	Na (mg/kg ± SD)	K (mg/kg ± SD)	Mg (mg/kg ± SD)	Fe (mg/kg ± SD)	Cu (mg/kg ± SD)	Zn (mg/kg ± SD)
Bacon	Turopolje pig	194.51 ± 93.82	0.07 ± 0.06	11123.21 ± 3138.49	6073.34 ± 5696.30	240.14 ± 140.27	26.65 ± 15.72	1.46 ± 1.01	19.42 ± 10.23
	Black slavonian pig	266.85 ± 130.12	0.07 ± 0.03	10781.88 ± 5499.04	2751.08 ± 2186.72	286.46 ± 122.09	16.48 ± 5.69	1.36 ± 0.68	24.60 ± 9.53
	Banijska šara	227.15 ± 52.62	0.17 ± 0.08	18743.70 ± 11237.54	4721.20 ± 3459.01	320.09 ± 133.19	50.99 ± 46.26	1.40 ± 0.58	31.03 ± 10.97
Dry-cured ham	Turopolje pig	226.69 ± 56.14	0.14 ± 0.07	20120.79 ± 11033.32	6508.25 ± 3928.45	283.75 ± 100.36	30.25 ± 12.38	1.22 ± 0.61	32.19 ± 14.00
	Black slavonian pig	260.41 ± 67.75	0.13 ± 0.01	25755.47 ± 7878.73	4149.33 ± 1636.09	307.65 ± 70.11	14.52 ± 2.10	1.23 ± 0.34	30.35 ± 8.94
	Banijska šara	238.30 ± 48.81	0.20 ± 0.06	26557.78 ± 15049.95	5682.53 ± 1379.08	307.41 ± 130.94	36.71 ± 15.91	0.70 ± 0.44	28.15 ± 11.35
Dry-fermented sausages	Turopolje pig	307.78 ± 72.64	0.25 ± 0.11	16997.47 ± 2803.34	7100.49 ± 1528.53	429.42 ± 135.52	37.30 ± 12.73 ^b^	1.69 ± 0.37	48.00 ± 10.24
	Black slavonian pig	293.66 ± 48.29	0.25 ± 0.02	14135.56 ± 1199.76	5855.11 ± 851.01	331.89 ± 119.10	19.48 ± 5.20 ^a^	1.58 ± 0.22	40.03 ± 11.43
	Banijska šara	283.80 ± 73.82	0.26 ± 0.02	16371.09 ± 4085.82	6771.06 ± 882.38	391.52 ± 60.70	40.28 ± 15.22 ^b^	1.08 ± 0.80	40.28 ± 12.73

Ca—calcium; P—phosphorus; Na—sodium; K—potassium; Mg—magnesium; Fe—iron; Cu—copper; Zn—zinc; ^a, b^ values within the same column under the same type of products (bacon, dry-cured ham, dry-fermented sausages) and with no common superscript differ significantly (*p* < 0.05).

**Table 2 foods-13-04175-t002:** Fatty acid profile in bacon, dry-cured ham, and dry-fermented sausages originating from three Croatian indigenous pig breeds.

Fatty Acid (%)	Bacon			Dry-Cured Ham			Dry-Fermented Sausages		
	Turopolje Pig	Black Slavonian Pig	Banijska šara	Turopolje Pig	Black Slavonian Pig	Banijska šara	Turopolje Pig	Black Slavonian Pig	Banijska šara
C14:0	1.35 ± 0.15	1.45 ± 0.1	1.33 ± 0.14	1.63 ± 0.39	1.26 ± 0.07	1.28 ± 0.06	1.38 ± 0.19	1.3 ± 0.06	1.26 ± 0.11
PALMITIC ACID (C16:0)	24.9 ± 1.26	23.78 ± 1.79	24.01 ± 1.72	28.15 ± 7.27	23.72 ± 1.36	24.27 ± 1.21	25.39 ± 0.51 ^b^	24.51 ± 1.09 ^ab^	23.95 ± 1.08 ^a^
C16:1n7*t*	0.39 ± 0.08	0.41 ± 0.13	0.45 ± 0.1	0.45 ± 0.11	0.43 ± 0.02	0.42 ± 0.12	0.42 ± 0.07	0.43 ± 0.1	0.44 ± 0.08
C16:1n7*c*	3.20 ± 0.27	3.24 ± 0.4	3.41 ± 0.83	3.82 ± 0.53	2.67 ± 0.53	3.52 ± 0.66	3.63 ± 0.50 ^b^	2.65 ± 0.39 ^a^	2.99 ± 0.43 ^ab^
C17:0	0.42 ± 0.12	0.44 ± 0.05	0.4 ± 0.13	0.50 ± 0.26	0.85 ± 0.46	0.32 ± 0.03	0.43 ± 0.14	0.46 ± 0.13	0.4 ± 0.04
C17:1	0.29 ± 0.10	0.3 ± 0.16	0.29 ± 0.09	0.37 ± 0.21	0.48 ± 0.25	0.23 ± 0.02	0.26 ± 0.07	0.27 ± 0.06	0.25 ± 0.03
C18:0	11.3 ± 1.28	10.39 ± 1.77	11.02 ± 1.11	12.81 ± 3.85	12.49 ± 1.11	11.37 ± 1.28	11.48 ± 0.78	12.83 ± 1.07	12.26 ± 0.86
C18:1n9*t*	0.17 ± 0.07	0.09 ± 0.08	0.13 ± 0.06	0.09 ± 0.10	0.32 ± 0.07	0.23 ± 0.02	0.16 ± 0.11	0.11 ± 0.1	0.22 ± 0.04
C18:1n9*c*	44.06 ± 2.62	35.57 ± 1.80	35.74 ± 1.61	36.61 ± 1.63	43.33 ± 2.45	35.14 ± 1.76	42.92 ± 1.35	36.91 ± 1.41	34.77 ± 1.56
C18:1n7	3.34 ± 0.27	12.67 ± 18.03	10.62 ± 15.47	4.22 ± 0.81	3.23 ± 0.55	13.23 ± 18.75	3.68 ± 0.35	8.55 ± 14.21	10.57 ± 16.21
C18:2n6*t*	0.16 ± 0.04	0.17 ± 0.05	0.17 ± 0.04	0.20 ± 0.03	0.15 ± 0.02	0.16 ± 0.02	0.16 ± 0.03	0.12 ± 0.05	0.14 ± 0.02
C18:2n6*c*	8.12 ± 1.21	8.55 ± 1.78	9.27 ± 2.34	8.22 ± 2.62	8.23 ± 2.78	7.13 ± 2.13	7.31 ± 0.73 ^a^	9.04 ± 1.55 ^ab^	9.58 ± 1.88 ^b^
C18:3n6	0.02 ± 0.03	n.d.	0.04 ± 0.03	0.02 ± 0.02	0.02 ± 0.03	0.04 ± 0.02	0.03 ± 0.03	n.d.	0.04 ± 0.04
C18:3n3	0.34 ± 0.15	0.32 ± 0.1	0.51 ± 0.19	0.32 ± 0.09	0.28 ± 0.15	0.34 ± 0.13	0.39 ± 0.11	0.4 ± 0.13	0.52 ± 0.15
C18:4n3	0.08 ± 0.03	0.07 ± 0.07	0.08 ± 0.05	0.03 ± 0.02	0.12 ± 0.08	0.05 ± 0.02	0.07 ± 0.03	0.08 ± 0.05	0.07 ± 0.01
C20:0	0.23 ± 0.05	0.22 ± 0.03	0.24 ± 0.03	0.22 ± 0.07	0.25 ± 0.07	0.2 ± 0.02	0.21 ± 0.03	0.22 ± 0.02	0.2 ± 0.03
C20:1n9	1.08 ± 0.16	1.28 ± 0.37	1.05 ± 0.21	1.24 ± 0.39	1.05 ± 0.17	0.96 ± 0.17	0.97 ± 0.10	1.02 ± 0.12	0.96 ± 0.14
C20:2n6	0.43 ± 0.07	0.49 ± 0.09	0.47 ± 0.14	0.45 ± 0.18	0.41 ± 0.13	0.34 ± 0.09	0.34 ± 0.03 ^a^	0.45 ± 0.07 ^b^	0.47 ± 0.11 ^b^
C20:3n6	0.05 ± 0.04	0.09 ± 0.02	0.04 ± 0.04	0.03 ± 0.04 ^a^	0.07 ± 0.04 ^b^	0.03 ± 0.04 ^a^	0.05 ± 0.04	0.08 ± 0.03	0.05 ± 0.05
C20:4n6	0.17 ± 0.06	0.18 ± 0.05	0.16 ± 0.03	0.16 ± 0.05	0.17 ± 0.1	0.15 ± 0.06	0.21 ± 0.04	0.24 ± 0.07	0.24 ± 0.1
C20:3n3	0.08 ± 0.04 ^ab^	0.04 ± 0.05 ^a^	0.13 ± 0.04 ^b^	0.05 ± 0.04 ^ab^	0.03 ± 0.04 ^a^	0.09 ± 0.03 ^b^	0.08 ± 0.03 ^ab^	0.07 ± 0.05 ^a^	0.13 ± 0.04 ^b^
C20:5 n3	0.01 ± 0.01	0.01 ± 0.02	0.04 ± 0.02	n.d.	n.d.	n.d.	n.d.	n.d.	n.d.
C23:0	0.02 ± 0.03	0.01 ± 0.02	0.04 ± 0.02	0.08 ± 0.06	0.15 ± 0.14	0.09 ± 0.07	0.05 ± 0.03	0.02 ± 0.04	0.04 ± 0.06

Results are expressed as the mean value in % of total fatty acids ±standard deviation; n.d.—not detected; ^a,b^ values within the same row under the same type of products (bacon, dry-cured ham, dry-fermented sausages) and with no common superscript differ significantly (*p* < 0.05).

**Table 3 foods-13-04175-t003:** Lipid quality indices of bacon, dry-cured ham, and dry-fermented sausages originating from three Croatian indigenous pig breeds.

	Bacon			Dry-Cured Ham			Dry-Fermented Sausages		
	Turopolje Pig	Black Slavonian Pig	Banijska Šara	Turopolje Pig	Black Slavonian Pig	Banijska Šara	Turopolje Pig	Black Slavonian Pig	Banijska Šara
IA	0.51 ± 0.05	0.46 ± 0.06	0.47 ± 0.06	0.72 ± 0.53	0.46 ± 0.05	0.48 ± 0.04	0.51 ± 0.02	0.49 ± 0.03	0.47 ± 0.04
IT	1.22 ± 0.13	1.08 ± 0.18	1.11 ± 0.13	1.69 ± 1.23	1.13 ± 0.16	1.16 ± 0.12	1.22 ± 0.05	1.23 ± 0.09	1.17 ± 0.09
HH	1.95 ± 0.22	1.90 ± 0.92	1.78 ± 0.72	1.68 ± 0.67	2.17 ± 0.26	1.67 ± 0.80	1.89 ± 0.07	1.82 ± 0.62	1.79 ± 0.68
PUFAs/SFAs	0.24 ± 0.05	0.29 ± 0.07	0.31 ± 0.09	0.22 ± 0.02	0.25 ± 0.09	0.22 ± 0.07	0.22 ± 0.02 ^a^	0.27 ± 0.05 ^ab^	0.29 ± 0.07 ^b^
TFAs (% of total FAs)	0.68 ± 0.15	0.66 ± 0.25	0.18 ± 0.17	0.74 ± 0.12	0.85 ± 0.15	0.81 ± 0.17	0.75 ± 0.17	0.65 ± 0.10	0.80 ± 0.10

Results are expressed as mean values ± SD; SD—standard deviation; SFAs—saturated fatty acids; PUFAs—polyunsaturated fatty acids; IA—atherogenic index; IT—thrombogenic index; HH—hypo-/hypercholesterolemic fatty acid ratio; TFA—trans fatty acids; ^a,b^ values within the same row under the same type of products (bacon, dry-cured ham, dry-fermented sausages) and with no common superscript differ significantly (*p* < 0.05).

## Data Availability

The data presented in this study are available on request from the corresponding author. The data are not publicly available due to privacy restrictions.

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
