# Peer review of "Comparison of the Nutritional Composition of Meat Products Derived from Croatian Indigenous Pig Breeds"

_foods, 2024, doi:10.3390/foods13244175_

Round 1
Reviewer 1 Report
Comments and Suggestions for Authors
The manuscript entitled „Comparison of the Nutritional Profiles of Meat Products Derived from Croatian Indigenous Pig Breeds” has great scientific potential.
Introduction
Page 2, lines 75-89: In my opinion this information should not be included in the introduction.
The authors rightly emphasize that eating a lot of meat is not desirable at the moment, but I really like their research arguments (Page 2, lines: 58-74). … consumers are increasingly focused on
the purchase of value-added meat products, which include traditional meat offerings,
products resulting from ecological farming, and those distinguished by health claims … I support the authors' opinion.
2. Materials and Methods
The methodologies have been described in great detail.
Figure 1. y axis: 100 g, not 100,00
90 g/100 g not 90,00
In my opinion, showing the sugar content is unnecessary.
Present statistically significant differences.
Table 1:
calcium content 566.62 ± 935.76 ??? please explain this
Zn content 31.46 ± 33.19?
46.04 ± 50.91
0.68 ± 0.82
111.46 ± 188.12
122.56 ± 191.31
Table 2:
C18:1n9c 4 - why are SD so high?
Author Response
We thank the Reviewer for his valuable and constructive comments. We have revised the manuscript and marked all changes in red.
Comments 1: Page 2, lines 75-89: In my opinion this information should not be included in the introduction.
Response 1: Thank you for pointing this out. We agree with this comment. Therefore we have moved this text to the Materials and Methods section (new subsection 2.2. Production of bacon, dry-cured ham, dry-fermented sausages).
Comments 2: Figure 1. y axis: 100 g, not 100,00, 90 g/100 g not 90,00, In my opinion, showing the sugar content is unnecessary. Present statistically significant differences.
Response 2: Thank you for pointing this out. We agree with these comments. Therefore we have made changes and corrections to the Figure 1, removing sugar content and decimal places on y axis and addding statistically significant differences.
Comments 3: Table 1: calcium content 566.62 ± 935.76 ??? please explain this, Zn content 31.46 ± 33.19?, 46.04 ± 50.91, 0.68 ± 0.82, 111.46 ± 188.12, 122.56 ± 191.31
Response 3: Thank you for pointing this out. We looked at all the data again and realised that the standard deviations are high due to the one value that stands out from the others in the group and outside the group. We therefore decided to consider this value as a statistical extreme and not to include it in the calculation. The data in the table has been changed accordingly.
Commnets 4: Table 2: C18:1n9c4 - why are SD so high?
Response 4: Thank you for pointing this out. There was a mistake in one sample decimal place. Now all numbers are once again checked.
Reviewer 2 Report
Comments and Suggestions for Authors
Dear Authors,
The sampling strategy outlined in the materials and methods section must be revised. The comparison of nutritional profiles among the three pig breeds is flawed due to the lack of control over critical variables, such as:
1. Feeding regimens: There is no evidence that the pigs were fed uniform diets or that their feed had comparable chemical compositions. Nutritional differences in the final products (proximate, mineral and fatty acid composition) could stem more from dietary variations than breed-specific traits.
2. Age and gender: The study does not account for the potential impact of the pigs' age or gender on meat composition. Such factors are well-documented as influencing nutritional properties.
3. Processing conditions: The meat products were produced under inconsistent conditions (e.g., spices, processing times, and curing methods). Without standardization, it is impossible to determine whether observed differences arise from breed characteristics or production methods.
4. The study needs more clarity on the specific production techniques used for each meat product. Variations in curing, fermentation, and other processing methods can significantly influence nutritional profiles. The absence of detailed information prevents readers from assessing whether these variables were adequately controlled or standardized.
Given these methodological issues, this manuscript is not suitable to be accept in its current form. The lack of control over critical variables makes it impossible to confidently attribute nutritional differences to pig breeds.
Author Response
We thank the Reviewer for his valuable and constructive comments. We have revised the manuscript and marked all changes in red.
Comments 1: Feeding regimens: There is no evidence that the pigs were fed uniform diets or that their feed had comparable chemical compositions. Nutritional differences in the final products (proximate, mineral and fatty acid composition) could stem more from dietary variations than breed-specific traits.
Response 1: Thank you for your valuable comment. We will try to clarify why it was not possible to use uniform diet.
Pigs indeed were not fed with the same feed, as each breed was bred in a different microgeographical location with some breeding particularities. The Black slavonian pig, for example, is traditionally fed with addition of oak nuts, which is not typical for the other two breeds. It is impossible to apply the same feeding regime to all three breeds, as each pig breed has its own peculiarities. We are aware of the possible effects of feeding, but if only one feeding regime were applied, the characteristics of breeding would be lost. This comment was also made by reviewer 1, who suggested that the aim of the study should be reformulated, so we have reformulated the aim according to the suggestions. We sinecirely hope that this clarified the lack of uniformed diet in the study.
Comments 2: Age and gender: The study does not account for the potential impact of the pigs' age or gender on meat composition. Such factors are well-documented as influencing nutritional properties.
Response 2: The information on age and sex (castrated male fattening pigs) has been added to the text (Subsection 2.2. Animals and meat products). The age of the pigs at slaughter was different for the different breeds, but this variable cannot be equated due to the different growth rate. The Banijska šara, for example, is never slaughtered at the same age as the Black Slavonian pig or the Turopolje pig. Therefore it is impossible to compare meat products that come from different pig breeds which were slaughtered at the same age.
Comments 3 and 4 combined: Processing conditions: The meat products were produced under inconsistent conditions (e.g., spices, processing times, and curing methods). Without standardization, it is impossible to determine whether observed differences arise from breed characteristics or production methods. The study needs more clarity on the specific production techniques used for each meat product. Variations in curing, fermentation, and other processing methods can significantly influence nutritional profiles. The absence of detailed information prevents readers from assessing whether these variables were adequately controlled or standardized.
Response 3 and 4: The processing conditions are now explained in more detail in subsection 2.2, as also requested by reviewer 3. The processing time and curing methods were the same for the same group of products. As for the spices, salt, black pepper and red paprika were used, which has now been included in the text. In relation to comment 3, we have included more information about curing and fermentation process in section 2.2.
Reviewer 3 Report
Comments and Suggestions for Authors
The paper submitted for review is a valuable work which concerns interesting issues associated with the quality of pork products derived from indigenous Croatian pig breeds. The experiment has been well planed. The Materials and Methods section is precise. The results have been clear presented and disscused. The conclusions from the research were correctly formulated. In my opinion, the manuscript could be suitable for publication after minor revisions.
Introduction
The research hypothesis should be indicate.
I suggest correcting the aim of the work into the following form: "The aim of this study was to characterize and compare the basic composition, mineral content and fatty acid (FA) profile of pork products (bacon, dry-cured ham and dry fermented sausages) derived from indigenous Croatian pig breeds. ".
Materials and Methods
There is no information about how the Authors measured salt, carbohydrates and sugars.
Line 153: AI is a standard abbreviation (artificial intelligence, AI). This should be changed.
Line 179: Please insert a dot.
Results and Discussion
Line 280: Please replace "is" with "was".
Line 285: Please move the last sentence to the next paragraph.
Line 302: Please replace "stud" with "study" and "concentration" with "content".
Lines 315-315: I suggest transforming the sentence into the following form: " Table 2. shows fatty acid profile of the analyzed products.".
Lines: 235-237, 338-339,347-348. Please note that the information is repeated.
Line 369: Please correct the reference.
Line 374: Please replace "SPA" with "SFA".
Lines 426-427: Please move the sentence to line 366.
Throughout the manuscript:
I suggest replacing the phrases "nutritional profiles" and "nutritional composition" used by the authors with the phrase "basic composition".
More references should be included in the text of the paper (lines: 61, 64, 66, 72, 84, 257, 269, 276, 282, 290, 293, 294, 402, 416).
Please use the abbreviations for the fatty acid groups that are explained the first time you use them.
Author Response
We thank the Reviewer for his valuable and constructive comments. We have revised the manuscript and marked all changes in red.
Comments 1: The research hypothesis should be indicate. I suggest correcting the aim of the work into the following form: "The aim of this study was to characterize and compare the basic composition, mineral content and fatty acid (FA) profile of pork products (bacon, dry-cured ham and dry fermented sausages) derived from indigenous Croatian pig breeds. ".
Response 1: Thank you for your comment. The aim of this study is now changed according to recomendation.
Comments 2: There is no information about how the Authors measured salt, carbohydrates and sugars.
Response 2: Thank you for pointing this out. Now information about determination of salt, carbohydrates and sugars are added at the end of section 2.4. The proximate analysis.
Comments 3: Line 153: AI is a standard abbreviation (artificial intelligence, AI). This should be changed.
Response 3: Thank you for pointing this out. Abbrevitation for atherogenicity index is now changed to IA and for thrombogenicity index is IT througt to whole manuscript.
Comments 4: Line 179: Please insert a dot.
Response 4: Thank you for your comment. Dot is now inserted.
Comments 5:Line 280: Please replace "is" with "was".
Response 5: Thank you for your comment. The replacement was made.
Comments 6:Line 285: Please move the last sentence to the next paragraph.
Response 6: Thank you for your comment. The sentance is now moved.
Comments 7:Line 302: Please replace "stud" with "study" and "concentration" with "content".
Response 7: Thank you for your comment. The replacements were made.
Comments 8: Lines 315-315: I suggest transforming the sentence into the following form: " Table 2. shows fatty acid profile of the analyzed products.".
Response 8: Thank you for your comment. The sentence is now tranformed according to suggestion.
Comments 9: Lines: 235-237, 338-339, 347-348. Please note that the information is repeated.
Response 9: Thank you for pointing this out. One sentance is now deleted.
Comments 10: Line 369: Please correct the reference.
Response 10: Thank you for pointing this out. The reference is now correct
Comments 11: Line 374: Please replace "SPA" with "SFA".
Response 11: Thank you for pointing this out. The replacement was made.
Comments 12: Lines 426-427: Please move the sentence to line 366.
Response 12: Thank you for your comment. The sentance is now moved.
Comments 13: I suggest replacing the phrases "nutritional profiles" and "nutritional composition" used by the authors with the phrase "basic composition".
Response 13: Thank you for your valuable comment but we prefer to use phrase "nutritional composition“ as several other authors i.e. reference 22.
Comments 14: More references should be included in the text of the paper (lines: 61, 64, 66, 72, 84, 257, 269, 276, 282, 290, 293, 294, 402, 416).
Response 14: Thank you for your comment. Several references was added to the text.
Comments 15: Please use the abbreviations for the fatty acid groups that are explained the first time you use them.
Response 15: Thank you for your comment.The abbreviations are now used through the text.
Reviewer 4 Report
Comments and Suggestions for Authors
Review of
Manuscript ID: foods-3330156-peer-review-v1
Type of manuscript: Article
Title: Comparison of the Nutritional Profiles of Meat Products Derived from Croatian Indigenous Pig Breeds
The aim of this study was to characterize and compare the nutritional composition, mineral content, and fatty acid profile of meat products derived from indigenous Croatian pig breeds. While it refers to specific breeds from Croatia, it can help the scientific community understand the peculiarities of indigenous breeds.
The paper is well-designed, conducted, and exposed. The English language is comprehensible and well-used. However, before recommending the paper for publication there are some issues that need to be addressed. Namely:
In Material and Methods, Meat products: regarding sausages, were always the same type of product? Refer to the specific product and its production description. While bacon and dry-cured ham can vary in the process, sausages can vary in ingredients besides the process.
Line 127 – Sixty should begin with a lowercase letter.
Line 198 – Remove “the” after regarding.
Line 199 – Include a paragraph to separate dry-cured hams from sausages the same way bacon is separated.
Lines 199-212 – It is important to know the ingredients and the production process of the sausages because it seems that the differences are not only due to the breed meat.
Line 225 (and many times throughout the text) – Include the names of the authors in “by the study from [21]”; “in the study by [xx]; write “by the study from Author [xx]”.
Line 231 – Rewrite “in Greek and Portuguese in Portuguese traditional sausages, respectively”. It’s confusing.
Line 234 – Correct “compering” to “comparing”
Lines 235-236 – Add Portugal.
Figure 1 – Include the significance of differences.
Lines 259-260 – Is there a logical reason for this “except for the bacon samples from black Slavonian pigs, where a content twice as high was found”? Please include it in the text.
There are better options for the Ca++ intake!
Lines 273-294 – Include references about all the minerals' functional properties.
Line 300 – Include the names of the authors [22]. Same as line 225.
Line 302 – Correct “stud” to “study”. Add the name of the author [7].
Lines 318-333 – Write the name of the fatty acids. Refer to their nomenclature and also their importance for consumers.
Line 369 – Is (Simopoulus, Bazarsadueva SV) a reference? Include the date. Add it to the references list
Lines 375- 415 – Include the names of the authors throughout the text.
Lines 426-427 – This sentence should be in the beginning of the section. Before the values are presented and discussed.
Line 454 – Which specific fatty acids? Refer to them.
Line 490 - Same as line 225, include the authors' names or write the text differently.
Author Response
We thank the Reviewer for his valuable and constructive comments. We have revised the manuscript and marked all changes in red.
Comments 1: In Material and Methods, Meat products: regarding sausages, were always the same type of product? Refer to the specific product and its production description. While bacon and dry-cured ham can vary in the process, sausages can vary in ingredients besides the process.
Response 1: Thank you for pointing this out. We agree with thise comment. Therefore we have added the detail explanation in the new Section - 2.2. Production of bacon, dry-cured ham, dry-fermented sausages.
Comments 2: Line 127 – Sixty should begin with a lowercase letter.
Response 2: Thank you for pointing this out. We agree with thise comment. The word Sixty was rewritten to sixty.
Comments 3: Line 198 – Remove “the” after regarding.
Response 3: Thank you for pointing this out. We agree with thise comment. The word „the“ was removed.
Comments 4: Line 199 – Include a paragraph to separate dry-cured hams from sausages the same way bacon is separated.
Response 4: Thank you for pointing this out. We agree with thise comment. The paragraph was included.
Comments 5: Lines 199-212 – It is important to know the ingredients and the production process of the sausages because it seems that the differences are not only due to the breed meat.
Response 5: Thank you for pointing this out. The additional text with the ingredients for making sausages has been added (Subsection 2.2.)
Comments 6: Line 225 (and many times throughout the text) – Include the names of the authors in “by the study from [21]”; “in the study by [xx]; write “by the study from Author [xx]”.
Response 6: Thank you for pointing this out. We agree with thise comment. The correction were made throught the whole text.
Comments 7: Line 231 – Rewrite “in Greek and Portuguese in Portuguese traditional sausages, respectively”. It’s confusing.
Response 7: Thank you for pointing this out. We agree with thise comment. The correction was made to : „in Greek and Portuguese traditional sausages, respectively“.
Comments 8: Line 234 – Correct “compering” to “comparing”
Response 8: Thank you for pointing this out. We agree with thise comment. The word was corrected.
Comments 9: Lines 235-236 – Add Portugal.
Response 9: Thank you for pointing this out. We agree with this comment. The word „Portugal“ was added.
Comments 10: Figure 1 – Include the significance of differences.
Response 10: Thank you for pointing this out. We agree with this comment. The significance of differences were added to the Figure 1.
Comments 11: Lines 259-260 – Is there a logical reason for this “except for the bacon samples from black Slavonian pigs, where a content twice as high was found”? Please include it in the text.
Response 11: This sentance is now deleted since according to other rewiever we reconsidered our data and concluded that one value can be observed as statistically extreme value so we didnt take it into account and now the different result is presented in the table.
Comments 12: There are better options for the Ca++ intake!
Response 12: Thank you for pointing this out. We agree with this comment. The correction was made to the text (word „is“ was changed to „could be“).
Comments 13: Lines 273-294 – Include references about all the minerals' functional properties.
Response 13: Thank you for pointing this out. We agree with this comment. The references were added to the text.
Comments 14: Line 300 – Include the names of the authors [22]. Same as line 225.
Response 14: Thank you for pointing this out. We agree with this comment. The correction were made throught the whole text as for Comment 6.
Comments 15: Line 302 – Correct “stud” to “study”. Add the name of the author [7].
Response 15: Thank you for pointing this out. We agree with this comment. The word „stud“ was corrected to „study“ and the name of the author was added.
Comments 16: Lines 318-333 – Write the name of the fatty acids. Refer to their nomenclature and also their importance for consumers.
Response 16: Thank you for pointing this out. We agree with this comment. The names of fatty acids were added as well as their importance to consumers.
Comments 17: Line 369 – Is (Simopoulus, Bazarsadueva SV) a reference? Include the date. Add it to the references list
Response 17: Thank you for pointing this out. We agree with this comment. The reference numbers were added to the text. These references were already on the list.
Comments 18: Lines 375- 415 – Include the names of the authors throughout the text.
Response 18: Thank you for pointing this out. We agree with this comment. The correction were made throught the whole text as for Comment 6 and Comment 14.
Comments 19: Lines 426-427 – This sentence should be in the beginning of the section. Before the values are presented and discussed.
Response 19: Thank you for pointing this out. We agree with this comment. The sentance was moved to the next pharagraph because it is related with the lipid quality indices.
Comments 20: Line 454 – Which specific fatty acids? Refer to them.
Response 20: Thank you for pointing this out. We agree with this comment. The fatty acid were added to the text.
Comments 21: Line 490 - Same as line 225, include the authors' names or write the text differently.
Response 21: Thank you for pointing this out. We agree with thise comment. The correction were made throught the whole text as for Comment 6.
Round 2
Reviewer 2 Report
Comments and Suggestions for Authors
Dear Authors,
Thank you for addressing the comments. Below is the feedback on your replies.
Comment 1: Feeding Regimens
I appreciate your explanation of traditional feeding practices and the reformulation of the study's aim. I understand the challenges in standardizing feeding regimens due to breed-specific practices and microgeographical variations. However, please expand the discussion in the manuscript regarding how these feeding differences may have influenced the meat products' chemical composition and quality attributes. Could you elaborate on any correlations between specific dietary components (e.g., oak nuts for Black Slavonian pigs) and the observed nutritional or sensory traits?
Comment 2: Age and Gender
I have a similar suggestion: Please include a more in-depth discussion in the manuscript about how age and gender may influence meat products' chemical composition and quality attributes. For instance, how might variations in slaughter age have affected the proximate composition or fatty acid profiles observed in your study?
Comments 3 and 4: Processing Conditions
The additional details on processing times, curing methods, and spice usage in Section 2.2 are helpful and address some of the concerns regarding standardization. Additionally, please discuss how processing methods (e.g., curing and fermentation conditions) could have influenced the final products' chemical composition and quality attributes. Could any observed differences between breeds be partially attributed to these variables?
Author Response
We would like to thank the Reviwer for his valuable and constructive comments. We have added the additional discussion (lines 505-527), taking into account and combining all three suggestions (Comments 1-4). The new text is marked in blue.